# Crosslinked Hyaluronic Acid Gels for the Prevention of Intrauterine Adhesions after a Hysteroscopic Myomectomy in Women with Submucosal Myomas: A Prospective, Randomized, Controlled Trial

**DOI:** 10.3390/life10050067

**Published:** 2020-05-15

**Authors:** Chen-Yu Huang, Wen-Hsun Chang, Min Cheng, Hsin-Yi Huang, Huann-Cheng Horng, Yi-Jen Chen, Wen-Ling Lee, Peng-Hui Wang

**Affiliations:** 1Department of Obstetrics and Gynecology, Taipei Veterans General Hospital, Taipei 112, Taiwan; eu.huang501@gmail.com (C.-Y.H.); whchang@vghtpe.gov.tw (W.-H.C.); alchemist791025@gmail.com (M.C.); hchorng@vghtpe.gov.tw (H.-C.H.); chenyj@vghtpe.gov.tw (Y.-J.C.); 2Institute of Clinical Medicine, National Yang-Ming University, Taipei 112, Taiwan; 3Department of Obstetrics and Gynecology, National Yang-Ming University, Taipei 112, Taiwan; 4Department of Nursing, Taipei Veterans General Hospital, Taipei 112, Taiwan; 5Biostatics Task Force, Taipei Veterans General Hospital, Taipei 112, Taiwan; sweethsin509@gmail.com; 6Department of Medicine, Cheng-Hsin General Hospital, Taipei 112, Taiwan; 7Department of Nursing, Oriental Institute of Technology, New Taipei City 220, Taiwan; 8Female Cancer Foundation, Taipei 104, Taiwan; 9Department of Medical Research, China Medical University Hospital, Taichung 404, Taiwan

**Keywords:** anti-adhesive gel, hyaluronic acid, hysteroscopic myomectomy, intrauterine adhesion, prevention, reduction

## Abstract

Intrauterine adhesion (IUA), fibrosis, and scarring resulting from damage to the endometrium is a rare but serious clinical disease, contributing to a significant impairment of reproductive function. Uterine instrumentation, especially that of a hysteroscopic myomectomy, has become the main cause of IUA. Therefore, a prospective randomized controlled study to assess the effectiveness and short-term safety of the use of hyaluronic acid gels in the prevention of IUA after a hysteroscopic myomectomy and an evaluation of the characteristics of IUA observed at follow-up are presented here. A total of 70 patients were analyzed at the end of 16 March 2020. The results show that the incidence of IUA in women who underwent a hysteroscopic myomectomy is 21.4% (15/70), overall. Women treated with hyaluronic acid gels have a statistically significantly lower incidence of IUAs than non-treated women (12.8% vs. 39.1%, *p* = 0.012). In addition, women in the anti-adhesive gel treatment group had a dramatically reduced severity of IUA than women in the no-treatment group (*p* = 0.002). Further analysis shows that the International Federation of Gynecology and Obstetrics (FIGO) classification type and the use of anti-adhesive gels are independent factors associated with moderate and severe degrees of IUA formation. The results here highlight the significant therapeutic benefits of the application of hyaluronic acid gels in women undergoing a hysteroscopic myomectomy, especially for those patients with a uterine myoma classified as FIGO type 2. Since the risk of IUA after a hysteroscopic myomectomy is high, especially for patients who have not received prophylactic anti-adhesive gels, the application of hyaluronic acid gels as a prevention strategy is highly recommended. More studies are encouraged to confirm our observation.

## 1. Introduction

Dr. Heinrich Fritsch reported the first case of post-traumatic intrauterine adhesion (IUA) in 1894 [1]. Subsequently, Dr. Joseph Asherman published a series of papers to describe this disease comprehensively, resulting in IUA being named Asherman syndrome or Asherman’s syndrome [2]. A large number of cases describing IUA have been reported in the literature [3,4,5,6,7,8,9]; however, IUA is often overlooked and considered a rare disease, partly because of a great variation in clinical features and vague symptoms and partly because of neglect and a failure to diagnose this disease, even for patients that present clear symptoms [4,5,6,7,8], resulting in an uncertain incidence or prevalence [10,11,12,13,14,15,16,17,18,19,20,21,22,23]. It is reported to be as low as 0.2% as an incidental finding in women treated with an intrauterine device insertion without gynecologic symptoms or signs and as high as more than 40% in women undergoing a hysteroscopic myomectomy procedure [10,11,12,13,14,15,16,17,18,19,20,21,22,23].

Normally, the repairing of the endometrium is tightly controlled by the orchestration of female sex hormones to coordinate the sequential and overlapping phases, including the hemostasis/inflammatory phase, proliferative phase, and remodeling phase in order to complete its cyclic change [24,25,26,27,28]. All are important for the scar-free healing of the endometrium and maintenance of the integrity of the endometrium after trauma. When extensive damage to the endometrium occurs, especially in certain conditions with trauma deep in the basal layer, the endometrial lining might be partially or totally absent, resulting in the approximation and fusion of surfaces on the opposite uterine wall. As a result, IUA develops. 

There are many predisposing and causal factor-associated injuries that are involved in the basal layer of the endometrium, although it is sometimes hard to show a clear and definable causative event [8]. For example, 3–4% of patients, after a cesarean section or undergoing postpartum dilation and curettage (D and C), may have a subsequent development of IUA [5]. More than one-third of patients with late spontaneous abortion D and C have a diagnosis of IUA [5]. These findings suggest that intrauterine instrumentation is associated with the greatest risk of developing IUA. The advancement of technology in hysteroscopic surgeries allows surgeons to manage nearly all intrauterine benign lesions with more convenience and confidence. As a result, surgeons will perform both a greater number of surgeries and increasingly complex intrauterine surgeries, such as hysteroscopic myomectomies for multiple myomas or the resection of apposing fibroids. All of these factors contribute to an increased number of IUA patients reported. Although the clinical features of IUA vary greatly, as shown above, there is no doubt that IUA is still largely idiosyncratic in many women because of frequent and major long-term complications, with an incidence ranging from 31% to 78% [5,12,13,14,25]. All efforts should be made to minimize the risk of the development of this troublesome disease.

The recognition of the importance of IUA development, especially that which occurs in patients after a hysteroscopic myomectomy, points to the urgent need to prevent or reduce its occurrence. In addition, subsequent surgery (intrauterine adhesiolysis) in women with IUA is more difficult and often takes a longer time [29,30]. Furthermore, it is frequently associated with a considerably higher complication rate [29,30]. After intrauterine adhesiolysis, many women still have a persistent IUA or have a recurrent disease [29,30]. 

A barrier is one of the frequently used tools in the prevention of adhesion, since, in theory, the separation of two opposite sides with a rough surface can prevent the contact of both sides, subsequently decreasing the risk of adhesion between the two sides. A barrier can be achieved via two strategies, and one is via an agent and the other is using a physical or mechanical barrier. Agents act as a barrier, including solid form, liquid form (hydroflotation agents), and gel form agents, and the components include polyethylene oxide–sodium carboxymethylcellulose gels and crosslinked hyaluronic acid (CHA) gels [9,17,29,30,31,32,33,34,35,36]. Physical or mechanical barriers include intrauterine suitable balloon catheters, Foley balloon catheters, Malecot catheters, silicone sheets, and intrauterine devices (IUDs) [8,10,17,36,37,38,39,40]. 

In fact, one study favored the use of mechanical methods, which might be much more effective than the use of a hyaluronic acid gel in the prevention of intrauterine adhesion reformation based on a greater reduction in the adhesion score [36]. Dr. Lin further found that the efficacy of the balloon was greater than that of the IUD [36]. However, there is no doubt that some patients might be very concerned about the application of a foreign object as a mechanical barrier for the primary prevention of IUA. The disadvantages of mechanical barriers include the need for further management, inconvenience, discomfort, the potential risk of infection, and anxiety, and all of these factors might influence their acceptability. By contrast, gels might be one of most convenient agents for application into limited and irregular spaces such as the intrauterine space, and most gel agents commonly include derivatives of hyaluronic acid (HA), with main components such as sodium D-glucuronate and N-acetyl-glucosamine, which is a linear polysaccharide with 25,000 repeating disaccharide units and composed of major supportive and protective components in the vitreous body, synovial fluid, cartilage, skin, and umbilical cord [29]. Earlier reports have shown that the application of a CHA gel reduced the severity of postoperative IUA after hysteroscopic procedures [29,30]. A meta-analysis also has shown that hyaluronic acid gel may reduce the incidence of moderate and severe IUA and may improve pregnancy rates after a miscarriage in patients (394 IUA patients for hysteroscopic adhesiolysis from six studies [29,36,41,42,43,44]) [45]. Furthermore, another meta-analysis, which was conducted to evaluate the effectiveness of the use of a HA gel to prevent IUA after a miscarriage [46], showed that hyaluronic acid gel has a protective role in the reduction of the recurrence rate of IUA (in a total of 625 patients from four studies [47,48,49,50]). Moreover, the most recent meta-analysis evaluated the efficacy of hyaluronic acid gel in preventing IUA following an intrauterine operation in 952 patients from seven randomized trials [29,30,42,47,48,49,50], where the results show that the use of a hyaluronic acid gel not only significantly reduced the incidence of IUA but also statistically significantly decreased the score for IUA after an intrauterine operation, suggesting that the protective role of using hyaluronic acid against the development of IUA is not affected by either the disease or the type of intrauterine surgery [51].

Besides the positive impact of using anti-adhesive agents for the prevention of the development of IUA, a follow-up hysteroscopy, either for examination or for treatment, has also been suggested for women after hysteroscopic surgeries [12,13,52,53].

Based on the aforementioned reasons, both strategies seem to contribute to a reduced risk of developing IUA after a hysteroscopic myomectomy. It is also well-accepted that strategies for primary prevention are much more effective than those for secondary prevention. Since there is still an absence of a consensus to recommend the routine use of anti-adhesive agents after complicated surgery and also because performing a follow-up hysteroscopy after hysteroscopic surgeries is not well-accepted, further evidence to support the beneficial effects of the application of anti-adhesive agents is required. To clarify the effectiveness of the use of anti-adhesive agents for the prevention of postoperative adhesion after fertility-preserved benign gynecological surgeries, we conducted a prospective randomized controlled study here to assess the effect and short-term safety of said agents. In addition, we also evaluated the characteristics of the postoperative adhesion when observed at a follow-up hysteroscopy.

## 2. Materials and Methods

### 2.1. Study Design and Participants 

This prospective randomized controlled trial (RCT) started in 25 April 2018, at one university-based tertiary medical center. This study protocol was approved by the Institutional Review Board, Taipei Veterans General Hospital (IRB number 2017-11-004B), and the study was conducted according to the guidelines of the 1975 Declaration of Helsinki for human experimentation. This trial was registered at the clinical trials government website (ClinicalTrials.gov ID: NCT04063085) in the following year (2019). The reason for this time lag between the approval document from the Taipei Veterans General Hospital and the registration document from the website is that the former included all patients who had a wish to join, regardless of the performance of the second-look operation. By contrast, the latter included only patients who underwent the second-look operation. The study protocol attempted to use different anti-adhesive agents, including auto-crosslinked hyaluronic acid (CHA, Hyalobarrier^®^ gel, Baxter, Pisa, Italy), crosslinked hyaluronic acid platform (CHA-P) gel (PROTAHERE absorbable adhesion barrier^®^, SciVision Biotech Inc., Kaohsiung, Taiwan), and a Seprafilm adhesion barrier^®^ (Genzyme Biosurgery, GENZYME Corp., Cambridge, MA, USA) for the reduction of the subsequent development of adhesion in women of childbearing age who require different kinds of fertility-preserving gynecological surgery for benign diseases, regardless of which surgical approach (laparoscopy, exploratory laparotomy, or hysteroscopy) was carried out as compared to no procedure in women with the same criteria. 

Based on the fundamental differences between the three types of surgery (exploratory laparotomy, laparoscopy, and hysteroscopy), the trial was further separated into three independent subgroups based on these three different surgeries. A Seprafilm adhesion barrier^®^ was only applied in the exploratory laparotomy subgroup. The report presented here features a subgroup analysis, which focused on patients in the hysteroscopic surgery subgroup. Although the results of this interim analysis revealed a trend favoring the use of an anti-adhesive agent gel for the reduction of developing postoperative adhesion, it was only applicable in the hysteroscopic myomectomy subgroup. Additionally, the trial conducted on the other two subgroups is still ongoing. Therefore, the effects of the use of anti-adhesive agents regarding the same questions in the other two subgroups are still uncertain. The detailed information of this hysteroscopic myomectomy subgroup analysis is given below. 

All patients aged between 20 and 65 years with diagnosed submucosal fibroids who planned to undergo a hysteroscopic myomectomy were screened for enrollment. Patients were excluded if there were medical or other serious conditions (an ongoing pregnancy was also included here) that may have interfered with either compliance or their ability to complete the study protocol or if they had a known allergy to modified hyaluronic acid. Once a patient met the eligibility criteria, the subject was offered enrollment in the current study. Upon providing consent, the participants were candidates for the randomization process. 

The aim of this subgroup analysis was to investigate the effectiveness of an anti-adhesive agent containing hyaluronic acid gel for the reduction of IUA development after a hysteroscopic myomectomy. The classifications of submucosal myomas (fibroids) were limited to the International Federation of Gynecology and Obstetrics (FIGO) classification types 0, 1, and 2 [54]. Since these patients have menorrhagia problems, all patients had at least one D and C or endometrial sampling before enrollment. To avoid potential bias from surgeons, participant randomization and grouping was only performed at the end of the hysteroscopic myomectomy procedure. Finally, the patients were randomly assigned to be treated with 10 mL of an anti-adhesive agent gel containing 40 mg/mL CHA-P gel (PROTAHERE absorbable adhesion barrier^®^, SciVision Biotech Inc., Kaohsiung, Taiwan) or 10 mL containing 30 mg/mL CHA gel (Hyalobarrier^®^ gel, Baxter, Pisa, Italy) for treatment (the treatment group); otherwise, the patients were assigned to the no-anti-adhesive agent gel treatment group (no-treatment group). Assignment was carried out at a 1:1:1 ratio for each group. Randomization was carried out as permuted blocks of 10 patients with a randomization code generated using software (SAS version 9.3; SAS Institute, Cary, NC, USA). The patients and the independent observer (a research nurse practice evaluating IUA and recording data) were unaware of the group allocation of the patients under examination during the follow-up. 

### 2.2. Procedures 

Women were scheduled for a hysteroscopic myomectomy in accordance with the procedural standards of the Taipei Veterans General Hospital, Taipei, Taiwan. The hysteroscopic myomectomy was performed under general anesthesia for all patients. All women were treated with intravaginal prostaglandin E1 analogs (two tablets of misoprostol, 200 µg), which were placed inside the posterior fornix of the vagina for two continuous nights before surgery to induce cervical softening. All surgeries were performed at 8 a.m. by three senior surgeons (Dr. Horng, Dr. Chen, and Dr. Wang). A preoperative prophylactic antibiotic with 1 gm of cefazolin sodium was intravenously given to all patients. The patients were lying in a lithotomy position, and the cervix was dilated under transabdominal ultrasound guidance with a Hegar dilator up to size 12. The Hegar dilator was also used to measure the length of the uterine cavity. An 8.5 mm working element alone with the outer sheath, 8.0 mm inner sheath, and 4 mm 12-degree telescope (Olympus, Germany) equipped with a hysteroscopic bipolar loop was inserted into the uterine cavity. The distending medium was isotonic normal saline (0.9% sodium chloride), which was introduced into the uterine cavity by an automated hysteroscopic distension pump with a set distension pressure of 100 mmHg. 

At the end of the hysteroscopic myomectomy, suction was applied to remove the debris and/or blood clots or accumulated fluid. Then, based on the randomization code, either nothing was applied to the uterine cavity in women who were randomized to the no anti-adhesive agent gel treatment group at the end of the procedure or, alternatively, 10 mL of CHA-P gel or 10 mL of CHA was inserted into the uterine cavity through a 30 cm sterile cannula in women who were assigned to the anti-adhesive agent gel treatment group. 

A follow-up operative hysteroscopy was scheduled for 12 weeks (±1 week) after the hysteroscopic myomectomy to avoid the active menstruation period, and the participants were checked to confirm the absence of pregnancy before the follow-up hysteroscopy. Any participant with a positive pregnancy test would not undergo the follow-up operation. The preparations and operation procedures of the follow-up surgery were the same as in the initial operation. The independent observer (a research nurse practice), who was not aware of the treatments that the patients received, classified the types and characteristics of the intrauterine pathologies observed and reported any intraoperative and postoperative complications related to the hysteroscopic myomectomy procedures. Hysteroscopic adhesiolysis for IUAs or the removal of residual tumors (fibroids) during the second-look hysteroscopic surgery was performed routinely. At the end of the procedure, all the participants received a prophylactic anti-adhesive agent gel treatment, regardless of whether they were originally assigned into the treatment or no-treatment group. 

### 2.3. Outcomes 

The primary outcomes were the rate of IUA and the severity of IUA. A modified AFS (American Fertility Society) classification of IUA was applied to assess the severity of IUA (Table 1). The severity was classified as Stage I (mild, scoring ≤ 2), II (moderate, scoring between 3 and 7), or III (severe, scoring ≥ 8). Safety was evaluated based on the occurrence of complications and adverse events possibly related to the CHA-P gel or CHA gel application, and this was recorded.

### 2.4. Statistical Analysis 

We used SAS version 9.3 (SAS Institute, Cary, NC, USA) and Stata Statistical Software version 12.0 (Stata Corporation, College Station, TX, USA) for all analyses to account for all of the complex sample designs. The characteristics of the patients were analyzed via descriptive statistics and presented as the means and standard deviations or percentages. Pearson chi-square tests were used for the categorical variables. Logistic regression analysis was applied to determine which was the single most dominant factor associated with the development of modified AFS Stage II or III (moderate or severe) IUA formation in women after a hysteroscopic myomectomy. A *p*-value of less than 0.05 was regarded as statistically significant. 

## 3. Results

### 3.1. The Characteristics of the Study Subjects 

The CONSORT flow chart of the participants in the current report is given in Figure 1. Of the 78 patients who were enrolled into the study, 48 patients were randomly assigned to the anti-adhesive agent gel treatment group (24 patients were treated with the CHA-P gel at 10 mL, and 24 patients were treated with the CHA gel at 10 mL) and the other 23 patients were not treated with any anti-adhesive agent gel (no-treatment group). The reasons for the exclusion of patients (n = 7) included patients being treated for a myoma associated with another pathology, such as a polyp, adenomyosis, or others (n = 3), and patients declining to participate with/without any reason (n = 4). However, one patient (in the CHA gel 10 mL group) did not receive a follow-up hysteroscopy on schedule in the anti-adhesive agent gel treatment group due to a pandemic event, and finally, 70 patients were analyzed. 

The characteristics of the patients in both (the anti-adhesive agent gel treatment and the no- anti-adhesive agent gel treatment) groups, including their age, their body mass index (BMI), their previous obstetrics and uterine surgery history, the type of uterine fibroids, the size of the uterine fibroids, and the number of the uterine fibroids, as well as other background factors, were similar without any statistically significant differences (Table 2).

### 3.2. The Outcomes 

The follow-up hysteroscopy showed that 12.8% of the patients (n = 6) in the anti-adhesive agent gel treatment group had a development of IUA, while 39.1% did (n = 9) in the no-treatment group, denoting a statistically significant difference between the two groups (*p* = 0.012) (Table 3). When the IUA was detected according to the modified AFS scoring system, we found that the women in the anti-adhesive agent gel treatment group had a statistically significantly lower severity of IUA development than those women in the no-treatment group did (*p* = 0.002). In the women with the occurrence of IUA, the majority of the patients (83.3%) that underwent anti-adhesive gel treatment had a mild severity compared to 11.1% of the patients without anti-adhesive gel treatment. 

Upon further dissecting the effectiveness of the different types of anti-adhesive gel on the primary prevention of IUA development after a hysteroscopic myomectomy, the results show that there was no statistically significant difference in IUA formation between patients who had received the CHA-P gel and those who had received the CHA gel (*p* = 0.352). In addition, the severity of IUA between the treatment groups was also not statistically significantly different (*p* = 0.497) (Table 4). However, CHA-P seemed to have a tendency to reduce the severity of IUA when IUA did develop, based on neither moderate nor severe IUA being found in the patients treated with CHA-P, and one case of moderate severity of IUA being found in the CHA group. In agreement with the results shown in Table 3, the patients treated with either the CHA-P gel or the CHA-gel not only had lower risks for the development of IUA but also had a lower severity of IUA when IUA did develop as compared to the patients without any anti-adhesive gel treatment, with statistically significant differences among the three groups (Table 4).

### 3.3. The Precipitating Factors for the Development of Modified American Fertility Society Stage II–III (Moderate and Severe) Intrauterine Adhesion 

To identify the factors that contributed to modified AFS Stage II to III IUA formation in patients after a hysteroscopic myomectomy, univariate and multivariate logistical regression analyses were performed. Univariate logistical regression analysis revealed that an older patient age, a high BMI, the presence of a previous uterine surgery history (cesarean section was included), the presence of an abortion history, a high gravidity, a high parity, a FIGO classification type 2 uterine myoma, the presence of multiple submucosal myomas, and big-sized submucosal myomas all are associated with an increased risk of developing modified AFS Stage II to III IUA (Table 5). 

A further stepwise logistical regression analysis was performed to identify the dominant key factors, and the results showed that uterine myoma classified as FIGO type 2 and the application of postoperative anti-adhesive gels significantly influenced the development of modified AFS Stage II to III (moderate and severe) IUA in patients after a hysteroscopic myomectomy (Table 6).

### 3.4. The Adverse Events 

The adverse events were similar in both groups, including abdominal cramping pain (21.3% in the anti-adhesive agent gel treatment group vs. 26.1% in the no-treatment group) and postoperative vaginal bleeding and/or spotting for more than seven days (29.8% in the anti-adhesive agent gel treatment group vs. 43.5% in the no-treatment group). All patients did not have operation-related complications or treatment-related complications. Pain was relieved by oral non-steroid anti-inflammatory drugs.

## 4. Discussion

### 4.1. Main Findings 

The main findings of the current report show that rate of IUA was 13% in women treated with a hysteroscopic myomectomy and anti-adhesive agent compared to 39% in women treated with hysteroscopic myomectomy alone, contributing to a total occurrence rate of IUA of 21% in women after a hysteroscopic myomectomy, suggesting that the incidence of IUA in women after a hysteroscopic myomectomy procedure was indeed very high, especially for those patients who did not receive anti-adhesive gels. Furthermore, the severity of IUA was moderate to severe for nearly 90% of women that did not receive prophylactic anti-adhesive agent therapy. Among nine patients with IUA in the no-anti-adhesive agent gel treatment group, eight patients (75%) were classified as having a moderate to severe form. By contrast, although six patients had IUA development in the anti-adhesive agent gel treatment group, IUA classified as moderate severity occurred in only one patient among the total 47 patients. The effectiveness of the anti-adhesive gels in reducing the risk and severity of IUA was not influenced by the potential confounding factors—such as a history of obstetrics, abortion, and previous uterus surgery—and there was no statistically significant difference between the treatment and no-treatment groups, suggesting that both types of anti-adhesive gel are similarly effective in the reduction of IUA development and reducing the severity of IUA when IUA does occur.

The dominant precipitating factors associated with a higher risk of modified AFS Stage II to III (moderate and severe) IUA in patients after a hysteroscopic myomectomy included a submucosal myoma classified as FIGO type 2 and the application of anti-adhesive gels.

### 4.2. Comparing with Other Randomized Controlled Trials (RCT) Using Hyaluronic Acid for the Primary Prevention of the Development of Intrauterine Adhesion after Hysteroscopic Myomectomy

Although it is well known that the risk of IUA development in patients after a hysteroscopic myomectomy is high and, additionally, that much more severe IUA is found, studies focusing on the investigation of the effectiveness or efficacy of anti-adhesive gels for the therapy of patients with a submucosal myoma who have undergone a hysteroscopic myomectomy are scarce. Dr. De Iaco and colleagues were pioneers in focusing on this topic. Their prospective randomized controlled trial (RCT) in 2003 evaluated the effectiveness of the application of a hyaluronic acid gel on the prevention of IUA formation in women after a hysteroscopic myomectomy [55]. This small RCT, enrolling 40 patients (18 in the treatment group and 22 in the no-treatment group), did not find a statistically significant difference regarding de novo IUA formation after a 9-week follow-up between the two groups (28% in the treatment group vs. 32% in the no-treatment group) [55]. However, another RCT conducted by Guida et al., which enrolled 132 patients (67 patients in the hyaluronic acid group and 65 in the control group), showed that patients receiving hyaluronic acid treatment exhibited significantly reduced IUA formation as compared with those without hyaluronic acid treatment (10.44% vs. 26.15%, *p* < 0.05) [30]. Our study showed a similar result to Dr. Guida’s study. Patients treated with prophylactic hyaluronic acid gel after a hysteroscopic myomectomy had a statistically significant reduction in IUA formation when compared with patients without treatment (12.8% vs. 39.1%, *p* = 0.012). Furthermore, the benefits of the application of postoperative hyaluronic acid are not only limited to a reduced risk of IUA formation but also include a significantly reduced severity of IUA if IUA formation is not totally avoided. 

### 4.3. Precipitating Factors Associated with an Increased Risk of the Development of Modified American Fertility Society Stage II–III (Moderate and Severe) Intrauterine Adhesion

Uterine myomas classified as FIGO type 2 were closely associated with an increased risk of moderate and severe degrees of IUA formation. This is reasonable, since a greater extension of the uterine myoma into the myometrium requires more resection of the tumor, which is associated with an increased risk of damage to the endometrium and subsequent IUA formation. Other precipitating factors, such as age, BMI, abortion history, pregnancy history, previous uterine surgery, the number of uterine myomas, and the size or the diameter of the uterine myoma were all correlated with IUA formation in patients after a hysteroscopic myomectomy, but they were not dominant key factors when the FIGO classification of a uterine myoma and the application of anti-adhesive gels were adjusted for. 

### 4.4. The Strengths and Limitations

The current study has the following strengths. All patients had a pathologically confirmed submucosal myoma without other associated diseases. As shown above, all had at least one D and C or endometrial sampling before enrollment to exclude other non-myoma underlying diseases. There was an extremely low dropout rate (less than 2%). The randomization was conducted at the end of the hysteroscopic myomectomy. All patients were treated by experienced and senior doctors (H.-C.H., Y.-J.C., and P.-H.W.), suggesting that the study population was homogeneous. Finally, the potential confounding factors, such as age, abortion history, pregnancy history, and previous uterine surgery, were included in the current study to minimize the risk of biases. 

There are some limitations in the current study. Since parts of the current study are still ongoing, especially with the exploratory laparotomy and laparoscopy subgroups, we do not know if the beneficial effects of the use of anti-adhesive agents on the prevention or reduction of adhesion are also apparent in the other two subgroups. The results of the current study are only applicable for the patients treated with a hysteroscopic myomectomy. Furthermore, the sample size of the current study was small. Moreover, we did not evaluate the reproductive performance of these patients; therefore, we do not know if the beneficial role of using anti-adhesive gels is actually present in women who have a need of future pregnancy. We suppose that the concerns regarding the future reproductive function of the patients might not be strong in the current study, since the age of the study subjects in the current study was relatively old, with a mean age of 45 years, even though the evaluation of reproductive performance is important for patients in their long-term follow-up. 

## 5. Conclusions

Intrauterine adhesion is still one of the most challenging diseases, although it is often neglected. The risk of the development of IUA is relatively high in women undergoing a hysteroscopic myomectomy. Based on a significant reduction in both the incidence and severity of IUA in women after a hysteroscopic myomectomy, we highly recommended the routine use of anti-adhesive agents, such as hyaluronic acid gels, in women who are due to undergo a hysteroscopic myomectomy, especially in those patients with an uterine myoma classified as FIGO type 2. 

## Figures and Tables

**Figure 1 life-10-00067-f001:**
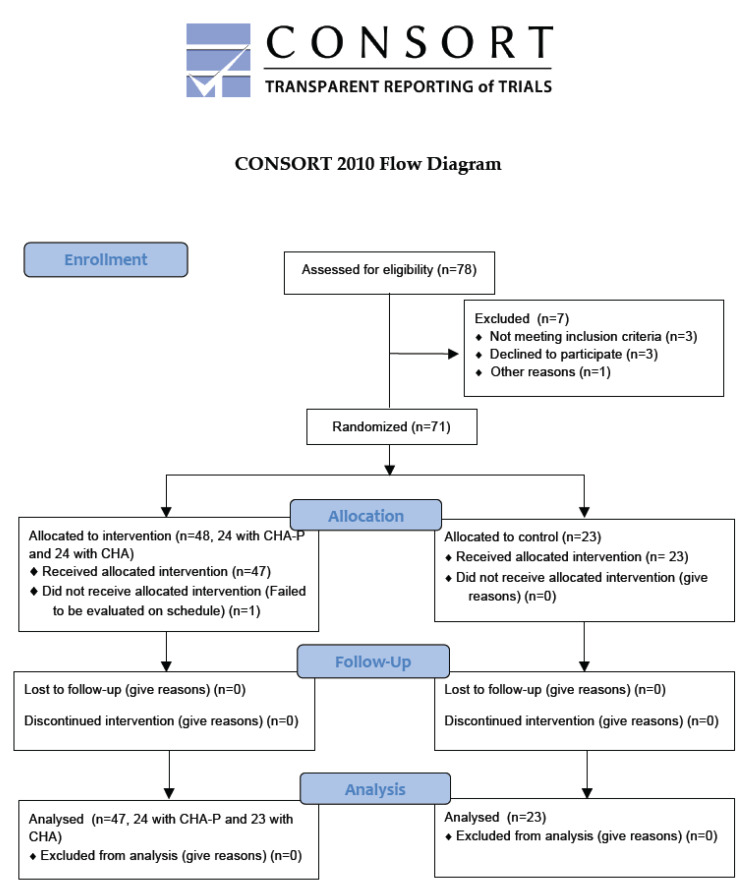
The CONSORT flow chart for the participants in the current report.

**Table 1 life-10-00067-t001:** Modified American Fertility Society grading method for intrauterine adhesion.

Scores	Description
1	Filmy adhesions with less than 1/3 enclosure
2	Filmy adhesions with 1/3 to 2/3 enclosure
4	Filmy adhesions with more than 2/3 enclosure
4	Dense adhesions with less than 1/3 enclosure
8	Dense adhesions with 1/3 to 2/3 enclosure
16	Dense adhesions with more than 2/3 enclosure

**Table 2 life-10-00067-t002:** Characteristics of the patients.

Characteristics		Treatment (n = 47)	No Treatment (n = 23)	*p* Value
**Characteristics of Patients**				
Age (years)		45.4 ± 3.6	45.2 ± 4.4	0.825
Body mass index (Kg/m^2^)		23.8 ± 3.6	22.7 ± 2.7	0.202
***Previous Uterine Surgery**				0.190
No		37 (78.7%)	21 (91.3%)	
Yes		10 (21.3%)	2 (8.7%)	
**Abortion History**				0.265
No		33 (70.2%)	19 (82.6%)	
Yes		14 (29.8%)	4 (17.4%)	
**Gravidity**		1.15 ± 0.98	0.78 ± 0.85	0.130
**Gravidity, times**				0.625
0		13 (27.7%)	10 (43.5%)	
1		19 (40.4%)	9 (39.1%)	
2		11 (23.4%)	3 (13.0%)	
3		3 (6.4%)	1 (4.3%)	
4		1 (2.1%)	0	
**Parity**		0.79 ± 0.72	0.61 ± 0.66	0.320
**Parity, times**				0.579
0		18 (38.3%)	11 (47.8%)	
1		21 (44.7%)	10 (43.5%)	
2		8 (17.0%)	2 (8.7%)	
**Characteristics of Myomas**				0.900
FIGO type	Type 1	32 (68.1%)	16 (69.6%)	
	Type 2	15 (31.9%)	7 (30.4%)	
Number		1.3 ± 0.5	1.3 ± 0.5	0.957
Maximal diameter (cm)		2.4 ± 0.5	2.2 ± 0.8	0.250
Maximal volume (cm^3^)		5.4 ± 3.2	4.1 ± 4.0	0.154

The data are presented as mean ± standard deviation or number (percentage). Treatment: the application of anti-adhesive hyaluronic acid gels. *Previous uterus surgery; cesarean section was included. FIGO: International Federation of Gynecology and Obstetrics.

**Table 3 life-10-00067-t003:** Findings of the follow-up hysteroscopy.

	Treatment (n = 47)	No Treatment (n = 23)	*p* Value
**Intrauterine Adhesion**			0.012
No	41 (87.2%)	14 (60.9%)	
Yes	6 (12.8%)	9 (39.1%)	
**Modified AFS Stage**			0.002
0	41 (87.2%)	14 (60.9%)	
I (mild)	5 (10.6%)	1 (4.3%)	
II (moderate)	1 (2.1%)	4 (17.4%)	
III (severe)	0	4 (17.4%)	

The data are presented as number (percentage). Treatment: the application of anti-adhesive hyaluronic acid gels. AFS: American Fertility Society.

**Table 4 life-10-00067-t004:** Findings of the follow-up hysteroscopy in patients treated with 10 mL containing 40 mg/mL crosslinked hyaluronic acid platform (CHA-P) gel, 10 mL containing 30 mg/mL crosslinked hyaluronic acid (CHA) gel, or no anti-adhesive gels.

	CHA-P Gel (n = 24)	CHA Gel (n = 23)	No (n = 23)	*p*-Value
**Intrauterine Adhesion**				0.031
No	22 (91.7%) ^a^	19 (82.6%) ^a^	14 (60.9%)	
Yes	2 (8.3%) ^a^	4 (17.4%) ^a^	9 (39.1%)	
**Modified AFS Stage**				0.014
0	22 (91.7%) ^b^	19 (82.6%) ^b^	14 (60.9%)	
I (mild)	2 (8.3%) ^b^	3 (13.0%) ^b^	1 (4.3%)	
II (moderate)	0 ^b^	1 (4.3%) ^b^	4 (17.4%)	
III (severe)	0 ^b^	0^b^	4 (17.4%)	

The data are presented as number (percentage). CHA-P (PROTAHERE absorbable adhesion barrier^®^, SciVision Biotech Inc., Kaohsiung, Taiwan); CHA gel (Hyalobarrier^®^ gel, Baxter, Pisa, Italy). No: no anti-adhesive agent gel treatment. AFS: American Fertility Society. ^a^ and ^b^: The comparison between the CHA-P gel and CHA gel (^a^: *p*-value = 0.352, ^b^: *p*-value = 0.497).

**Table 5 life-10-00067-t005:** Precipitating factors affecting the development of modified American Fertility Society Stage II to III (moderate and severe) intrauterine adhesion.

Characteristics	n	Odds Ratio (95% CI)	*p*-Value
**Characteristics of Patients**			
**Age (years)**			
<43.9	27	1 (Reference)	
≥43.9	43	3.03 (1.90–4.83)	<0.0001
**Body Mass Index (Kg/m^2^)**			
<23.22	32	1 (Reference)	
≥23.22	38	4.11 (2.38–7.13)	<0.0001
***Previous Uterine Surgery**			
No	58	1 (Reference)	
Yes	12	2.63 (1.81–3.80)	<0.0001
**Abortion History**			
No	52	1 (Reference)	
Yes	18	2.99 (1.97–4.53)	<0.0001
**Gravidity**			
<2	51	1 (Reference)	
≥2	19	2.99 (1.97–4.55)	<0.0001
**Parity**			
<2	60	1 (Reference)	
≥2	10	2.72 (1.87–3.96)	<0.0001
**Characteristics of Myoma**			
**FIGO Type**			
1	48	1 (Reference)	
2	22	3.75 (2.28–6.17)	<0.0001
**Number**			
1	49	1 (Reference)	
>1	21	3.32 (2.11–5.23)	<0.0001
**Maximal diameter**			
≤2.7 cm	51	1 (Reference)	
>2.7 cm	19	3.70 (2.27–6.02)	<0.0001
**Maximal volume**			
≤6.8 cm^3^	47	1 (Reference)	
>6.8 cm^3^	23	3.54 (2.20–5.72)	<0.0001
**Application of Postoperative Anti-Adhesive Gels**			
No	23	1 (Reference)	
Yes	47	0.25 (0.15–0.42)	<0.0001

n: number of patients; CI: confidence interval; *previous uterine surgery (cesarean section was included); FIGO: International Federation of Gynecology and Obstetrics.

**Table 6 life-10-00067-t006:** Stepwise logistical regression analysis of the significant variables in patients with and without modified American Fertility Society Stage II to III (moderate and severe) intrauterine adhesion.

	Odds Ratio (95% Confidence Interval)	*p*-Value
**FIGO type**		
1	1 (Reference)	
2	108.36 (8.95–1312.54)	<0.0001
**Application of Postoperative Anti-Adhesive Gels**		
No	1 (Reference)	
Yes	0.04 (0.00–0.33)	0.003

FIGO: International Federation of Gynecology and Obstetrics.

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
