# Peer review of "Crosslinked Hyaluronic Acid Gels for the Prevention of Intrauterine Adhesions after a Hysteroscopic Myomectomy in Women with Submucosal Myomas: A Prospective, Randomized, Controlled Trial"

_life, 2020, doi:10.3390/life10050067_

Round 1
Reviewer 1 Report
The authors significantly improved the manuscript according to previous comments
Author Response
Dear Reviewer:
Please read the attached file and your favourable consideration is much appreciated.

Reviewer 2 Report
Dear Authors,
this is an excellent piece of scientific work. Congratulations on the idea and the soundess of your research.
I made some comments in the text. The manuscript needs minor revison in case of the merit and very extensive language edition. I'm not native speaker myself, but it is difficult to follow your writting. Please try to write shorter sentences.
In general discussion is to long, please try to minimize it by 25%.

Author Response
Dear Reviewer
Thank you very much for your kindness and favourable consideration. Please read the attached file.

Reviewer 3 Report
The Authors conducted a prospective randomized controlled study on 70 women undergoing hysteroscopic myomectomy to assess the effectiveness and short-term safety of the use of hyaluronic acid gels in the prevention of intrauterine adhesion (IUA) after this surgery, and to evaluate the characteristics of IUA observed at follow-28 up.
Women treated with hyaluronic acid gels had a statically significantly lower incidence of the developing IUA than women without did (12.8% vs. 39.1%, p = 0.012). In addition, women in the anti-adhesive gel treatment group had a dramatically reduced severity of IUA than women in the no-treatment group (p = 0.002).
The topic is interesting, in fact, it has been the subject of several studies in recent years. The study is well conducted, but the number of patients is small, and the results are partial, referring only to intrauterine adhesion related to hysteroscopic myomectomy. The article is well written, but it does not add much to the results of previous studies. The article requires a major revision before accepting for publication.
- In the introduction, it would be appropriate to add a digression also on other methods, mentioned in literature, used to prevent the occurrence of IUA such as the intrauterine balloon or the intrauterine contraceptive (i.e. Xiaona Lin et al, A comparison of intrauterine balloon, intrauterine contraceptive device and hyaluronic acid gel in the prevention of adhesion reformation following hysteroscopic surgery for Asherman syndrome: a cohort study. European Journal of Obstetrics & Gynecology and Reproductive Biology, 2013).
- The total number of patients is too small. In particular, there are few controls. This is an important bias that could affect the results of the study. Why did the authors decide to randomly assign in a 2:1 ratio? Please explain
- Patients aged 20 to 65 years old were recruited in this study. Choosing this wide age group could be a source of bias because patients have a different hormonal status and endometrium characteristics according to the age. In addition, younger patients may have fewer surgical procedures on the uterus than older patients. It would be useful to report the rate of previous interventions on the uterus in the patient’s medical history which could be adjunctive risk factors for the appearance of IUA. Furthermore, it would be more appropriate to stratify the results according to the age of the patients and their personal medical history.
- In the discussion, the choice to include the meta-analyses and the RCTs in two distinct paragraphs creates a fragmentation in the article making it less accessible to the reader.
Author Response
Dear Reviewer
We appreciate your valuable comments and suggestion. We modified the article to fulfil your comments points by points. Please see the attached file. Your favourable consideration is much appreciated.

Round 2
Reviewer 3 Report
The authors have satisfactorily responded to all my questions and made the necessary changes to the manuscript. The article can be accepted for publication.
This manuscript is a resubmission of an earlier submission. The following is a list of the peer review reports and author responses from that submission.
Round 1
Reviewer 1 Report
This paper by Huang and colleagues has fundamental methodological problems.
An interim and subgroup analysis of an RCT should only be done when the interim results might have a major impact, for example a premature discontinuation of the study because of serious adverse events, etc. In this case, there are only 10 patients in each arm of the study, so there is nearly zero statistical power and therefore no meaningfull conclusion can be drawn.The data from this trial (NCT04063085) should be published in one paper, including this subanalysis on hysteroscopic myomectomies. In this publication the methodology will have to be described in full detail and can then be scrutinized. At first sight there are already several major methodological issues: study was started in 2018 and only registered in 2019, combination of different agents and different operations (hysteroscopy, laparoscopy, laparotomy), statistical power,....
The systematic review and meta-analysis of the literature on prevention of IUA post hysteroscopic myomectomy has limited or no scientific value. "You can't review what is not there". It is useless to perform a meta-analysis of 2 small RCTs. If we want to shed light on the use of anti-adhesions methods after hysteroscopic myomectomy, we need a large and high quality RCT similar to the one by Guida and colleagues, while addressing its shortcomings. Several prospective, multicentric studies on this topic are ongoing.
Reviewer 2 Report
the manuscript is well written
Major concern
This manuscript is too long and it would be better if it is extensively reduced. for example maximum cuts could prevail for the first part which is only theoritical knowledge
Right now it is much more like a book chapter
minor problem:
for ex in the abstract : there is no need for the reader to memorize the clinical trial number as abstract shoul generally be limited to 250 words
All abbreviations should carefully be rechecked for ex I could'nt find the abbreviation of HM , I guess it is hysteroscopivc myomectomy?
Figures should be reedited, words are hardly distinguishable in a standard paper print format